# Self-Assembled Monolayers of Push–Pull Chromophores as Active Layers and Their Applications

**DOI:** 10.3390/molecules29030559

**Published:** 2024-01-23

**Authors:** Junlong Wang, Virginie Gadenne, Lionel Patrone, Jean-Manuel Raimundo

**Affiliations:** 1Aix Marseille Univ, CNRS, CINaM, AMUTech, 13288 Marseille, France; junlong.wang@etu.univ-amu.fr; 2ISEN, Université de Toulon, Aix Marseille Univ, CNRS, IM2NP, AMUtech, 83041 Toulon ou Marseille, France; virginie.gadenne@yncrea.fr

**Keywords:** self-assembled monolayers, push–pull chromophores, active layers, optoelectronics

## Abstract

In recent decades, considerable attention has been focused on the design and development of surfaces with defined or tunable properties for a wide range of applications and fields. To this end, self-assembled monolayers (SAMs) of organic compounds offer a unique and straightforward route of modifying and engineering the surface properties of any substrate. Thus, alkane-based self-assembled monolayers constitute one of the most extensively studied organic thin-film nanomaterials, which have found wide applications in antifouling surfaces, the control of wettability or cell adhesion, sensors, optical devices, corrosion protection, and organic electronics, among many other applications, some of which have led to their technological transfer to industry. Nevertheless, recently, aromatic-based SAMs have gained importance as functional components, particularly in molecular electronics, bioelectronics, sensors, etc., due to their intrinsic electrical conductivity and optical properties, opening up new perspectives in these fields. However, some key issues affecting device performance still need to be resolved to ensure their full use and access to novel functionalities such as memory, sensors, or active layers in optoelectronic devices. In this context, we will present herein recent advances in π-conjugated systems-based self-assembled monolayers (e.g., push–pull chromophores) as active layers and their applications.

## 1. Introduction

Self-assembly is a ubiquitous phenomenon in nature, in which building units composed of atoms, (bio)molecules, polymers, colloids, or particles are capable of self-organizing into ordered and/or functional patterns or superstructures [1,2]. Self-assembling proceeds randomly or directionally and is governed by local interactions (repulsive or attractive forces) between the monomer units themselves, with or without external direction [3]. Nanoscale self-assembling occurs at interfacial or solution and constitutes the easiest bottom-up approach [4].

Among these self-assemblies, self-assembled monolayers, classically referred to as SAMs [5,6], constitute an interesting approach to surface functionalization, fine-tune the properties of a surface of interest, and are suitable for industry. The concept of SAMs was first introduced at the liquid–gas interface [7,8] prior to being reported in the late 40s by Bigelow et al. [9,10], then successively by Sagiv et al. [11] and Nuzzo et al. [12], on substrates and suggested later for applications by Whitesides et al. [13]. Due to the wide range of applications of SAMs, the pioneers of the field were awarded in 2022 by the Kavli prize in nanoscience [14]. SAMs are long-range ordered two-dimensional single-molecular layers of well-oriented chemisorbed or physiosorbed organic compounds, which assemble spontaneously on various surfaces in the gas phase or in solution. The compounds used for the fabrication of SAMs are generally amphiphilic molecules composed of three parts [15]: *(*i) the anchoring or head group, which interacts and binds to the surface; (ii) a spacer, typically a molecular backbone made of an aliphatic chain or aromatic oligomer imparting the molecular packing and order; (iii) the end or tail group, which is responsible for the surface energy, chemistry, and topography of the outer interface [16,17] (Figure 1).

Based on the molecular deposition techniques used, uniform homomolecular or heteromolecular SAMs can be obtained at the liquid/liquid, liquid/solid, air/liquid, or air/solid interfaces with the existence of two monolayer types, namely, the Gibbs and Langmuir monolayers [18,19]. However, the development and performance of such SAMs can be hampered by some hurdles. Indeed, the quality and properties of the SAMs depend on several parameters, among them, the thickness of the monolayer, the molecular orientation and order, the uniformity and coverage, the chemical composition, the odd–even effects of the linker [20], electrostatic effects [21], and the thermal and chemical stabilities [22]. In addition, several factors affect the SAM formation [23], i.e., temperature [24,25], immersion time [26], solvent [27,28], concentration, humidity, and O_2_ contents [29,30], and need to be mastered and controlled to attain the desired properties.

SAMs based on alkanethiols [14,31], alkanesilanes [32], alkanecarboxylates [33], alkanephosphonates [34], etc., on metallic or metal oxide [35] substrates have been extensively studied, both experimentally and theoretically, along with the development of novel efficient headgroups such as *N*-heterocyclic carbenes (NHCs) [36,37] or multidentate adsorbates [38]. Indeed, SAMs are versatile and inexpensive surface coatings that have been used in both static or dynamic ways (by using various stimuli) for a variety of applications including micro- and nanofabrication [39,40], sensors [41], batteries [42], biological [43], energy [44], and electronics [45], among many other applications.

Recently, SAMs made from π-conjugated systems have gained wide attention due to their intrinsic optical and electrical properties, which can be used in molecular electronic devices, molecular (bio)sensors, bioelectronics, photovoltaics, and so on [46].

For instance, in molecular electronics, these SAMs are commonly used to fine-tune the work functions of metallic or inorganic electrodes to minimize the energy barriers for holes, electron injection, or extraction from an active organic layer [47]. Such highly ordered π-conjugated chromophores are also often encountered in nature and act either as photon absorbers, electron donors and acceptors (chlorophyll, pheophytins, quinines, etc.), or as eye photoreceptors in human retinas [48], resulting in efficient photoinduced charge separation and electron transfer. Thereby, self-assembling π-conjugated chromophores constitute a key-point in organic nanodevices to improve their properties and operation [49]. The elaboration of SAMs made from organic π-conjugated molecules will follow the same recipes as described for alkane-based self-assembled monolayers considering additional parameters that could influence the SAMs, like the dipole effect [50,51], the alkyl spacer influence between the aromatic backbone and the surface [52], the number of aromatic rings, and so on.

P-conjugated systems are versatile organic materials for optoelectronics whose optical and electrical properties are strongly impacted when self-assembled as thin films [53] and need to be well controlled and oriented to optimize the performance of the devices in which they are implemented [54,55]. Among them, push–pull chromophores (referred to classically as donor–π–bridge-acceptors or D–π–A) are a class of peculiar importance [56] with various shapes, such as linear, branched, twisted, planar, nonplanar, etc. [57,58], exhibiting linear and/or nonlinear optical properties and electrical properties that are useful, for instance, in the design, conception, and manufacture of electronic devices. These molecules typically consist of an electron-donating (push) unit and an electron-withdrawing (pull) unit connected through a conjugated bridge (Figure 2).

The push and pull units create an electronic imbalance, leading to improved charge transfer and exciton separation, which is crucial for efficient optoelectronic devices. By carefully designing and modifying the molecular structure of push–pull chromophores, researchers can tailor their optoelectronic properties for specific applications, aiming for increased efficiency and performance in devices. Adjusting the strength of the donor and acceptor units can optimize the HOMO (highest occupied molecular orbital) and LUMO (lowest unoccupied molecular orbital) energy levels, affecting the absorption and emission properties. Extending the conjugated bridge increases the delocalization of electrons, facilitating better charge transport and reducing energy loss in the material. Introducing bulky groups strategically can affect the molecular packing, enhancing solid-state properties and preventing aggregation-induced quenching fluorescence. Incorporating electron-donating or electron-withdrawing substituents on the aromatic rings can fine-tune the electronic properties of the chromophore. Introducing solubilizing groups can enhance the material’s processability, making it easier to fabricate thin films or coatings for devices. Systematically modifying the molecular structure based on computational simulations and experimental results can lead to the identification of optimal structures with improved optoelectronic properties. However, in most cases, these push–pull chromophores are embedded and used in their final form in polymeric or composite matrixes (as doped or grafted, poled or not) [59] and rarely as SAMs. Thus, we will report herein some recent results and applications dealing with push–pull chromophore-based SAMs and their applications.

## 2. SAMs from Push–Pull Chromophores for Dye-Sensitized Solar Cell Applications

Push–pull chromophores have been extensively studied in recent years with the aim of incorporating them into the conception of organic solar cells [60], mainly as donor materials. Among various solar cell technologies, their use in dye-sensitized solar cells (DSSCs) has revealed interesting results since such donor–π–acceptor (D–π–A) sensitizers exhibit strong molar absorption, making it possible to use thin oxide films compatible with the improvement in the open circuit voltage [61,62,63,64,65]. The operating principle of DSSCs incorporating push–pull assemblies at inorganic oxide surfaces is depicted in Figure 3, extracted from the recent article by D’Annibale et al. [66]. Upon light absorption, an electron is transferred from a p-type semiconducting substrate (valence band, VB) to the donor group of the dye (D), which becomes excited (D*), and the acceptor moiety withdraws the electron from the donor through the spacer (step a). The latter both acts as an electron vehicle and is the seat of charge separation. At last, the acceptor donates the electron to the oxidized form of the redox mediator (step b).

For good DSSC operation, it is mandatory that the dyes be grafted onto the inorganic mesoporous semiconductor oxide film in controlled, organized stacks, enabling the determination of energy alignments and high charge-transfer kinetics to improve light harvesting [67]. Such grafting as SAMs can be carried out using push–pull with an adequate anchoring group [68] that could be profitably optimized [69]. As for incorporating such assemblies within DSSCs, one can cite, for example, the work of Gholamrezaie et al. [70,71], using π-conjugated quinquethiophene-derivative chromophores. Therefore, it is necessary to develop the energy-level engineering of chromophores on the metal oxide surface, which can be achieved by the right design of push–pull chromophores through the nature and length of the π-bridge groups, together with using various acceptor and donor groups with different electron affinities. Among the studies on various acceptors, one can cite the work of Keremane and co-workers [72] (Figure 4), Paul and Sarkar on PCBM-based acceptors (Phenyl-C61-butyric acid methyl ester) [73], and Mustafa et al. [74] on the theoretical study of the influence of the acceptor’s nature.

Donor nature has also been extensively studied, for instance, with auxiliary methoxy as a donor, to improve metal-free organic dye performance for DSSCs [75]. Some works have been devoted to the effect of both donors and π-spacers [76,77,78] or solely focused on the latter, for instance, with unconventional helical push–pull, enabling internal charge transfer and leading to good injection into the conduction band of TiO_2_ [79], or π-bridge extension to decrease the gap and widen the light absorption range [80]. Such spectral absorption broadening has also been achieved by the co-adsorption of two push–pull dyes [81]. Moreover, the donor–π-bridge–acceptor (D–π–A) structure represents a convenient configuration for a high charge separation rate on the organic sensitizer. In D–π–A molecules, intramolecular charge transfer (ICT) occurs efficiently between the donor and the acceptor parts, and the intramolecular electronic relaxation has been shown to play a role in the injection process [82]. Improvement in the intramolecular charge transfer rate is indeed a key issue to improve DSSC efficiency [83,84]. To fulfill the requirement of a broader spectral response, it is also possible to add an internal electron-withdrawing unit within some new donor–acceptor–π–acceptor (D–A–π–A) dyes by further grafting benzothiadiazole, benzotriazole, diketopyrrolopyrrole, or quinoxaline to the usual D–π–A structure. For instance, Demirak et al. [85] worked on novel unsymmetrical push–pull sensitizers based on triarylamine-substituted quinoxaline push–pull dyes with the aim of improving the performance of DSSCs. Furthermore, such a strategy can also be completed by the proper choice of side chains. Indeed, unsymmetrical push–pull porphyrazine has also been reported by Fernandez-Ariza et al. [86]. With the aim of improving the absorption features of porphyrins and phthalocyanines, the authors proposed a molecule with a phthalocyanine core incorporating a specific design of the peripheral substituents (Figure 5), thus enabling increased absorption in the red, improved solubility, and the possibility of tuning the electronic parameters. With such a seminal molecule, they obtained a PCE (power conversion efficiency) of 3.42% and opened the way for the improvement in panchromatic light harvesting by properly tuning the peripheral functions.

In particular, in a more recent work, the authors studied the effect of tuning of the electron-donor unit of push–pull porphyrazines [87] (Figure 6), showing its effect on both adsorption and electron injection processes, thus highlighting that porphyrazine design is a delicate process with a significant effect on the electronic properties of organic dyes and therefore on DSSC operation.

Variations on the push–pull structure have been studied on other several chromophores such as distyryl boron dipyrromethenes as near-infrared sensitizers [88], exhibiting a wide absorption range from UV-visible to near-infrared. Interestingly the authors also demonstrated that the size of the TiO_2_ nanoparticles is also a parameter to tune according to the molecular size of the dye to improve the DSSC performance. As seen with the porphyrazine example, macrocycle-based push–pull chromophores such as phthalocyanines [89,90,91] (Figure 7) and porphyrins [92,93,94] are good candidates for use as organic dyes within DSSCs and have been extensively studied during the last decades.

Concerning porphyrins, push–pull-type porphyrin-based dyes have shown the best results [95,96,97,98,99,100,101,102], and, notably, with such compounds, Grätzel et al. managed to obtain an efficiency of about 13% [103,104]. Panagiotakis et al. [105] have reported increased efficiency, with PCE ranging from 5 to 7.6%, using carefully designed zinc porphyrin push–pull derivatives grafted via cyanoacrylic acid on TiO_2_. Particularly, they showed, using a π-conjugated spacer between the chromophore and the anchoring group, enhanced electron transfer and hindered undesirable aggregation on the TiO_2_ surface. Cheema et al. [106] have obtained near-infrared absorption by conjugating the porphyrin to indolizine as a planar strong electron donor, thus inducing π–π interactions such as head-to-tail dye aggregation (Figure 8).

As for phthalocyanines, they exhibit very good photo- and electrochemical stability and high light-harvesting capability in the red/NIR (near-infrared) spectral regions [89]. Interestingly, the optoelectronic properties of phthalocyanines can be tuned through the proper choice of the organic substituents around the core, since they have a direct impact on the HOMO–LUMO energy levels as well as on the electron density, but also by the metalation nature of the core to reach long-living excited states. As is the case for porphyrin-based DSSCs, it is necessary to avoid phthalocyanine aggregation at the oxide surface. Because phthalocyanines easily exhibit aggregation on titanium surfaces, Milan et al. studied unsymmetrically substituted push–pull Zn phthalocyanines on original SnO_2_-based DSSCs [107].

## 3. SAMs of Push–Pull Chromophores to Improve Perovskite Solar Cell Performances

Beside DSSCs, which represent the main solar cell type incorporating push–pull self-assembly, recent works have also been reported with beneficial uses of push–pull chromophores within perovskite solar cells. Indeed, in addition to energy band alignment and improvement in electron transfer at the interface, such solar cells undergo a drastic lack of stability, and push–pull chromophores have been shown to be able to address all these critical issues. Liu and co-workers have designed an acceptor–donor–acceptor chromophore as an interfacial organic layer whose push–pull effect promotes the charge transfer between organic and inorganic layers in 2D perovskite solar cells by lowering the bandgap of the organic spacer of the perovskite [108]. The push–pull chromophore is made of dithienyl diketopyrrolopyrrole (DPP-2T) with two ammonium cations attached to both sides of the DPP unit, thus allowing hydrogen bonds with inorganic [PbI6]4- sheets within the perovskite cell. Incorporating such DPP-2T push–pull chromophores enables the improvement of the current (Figure 9) and the performance of the cell with a PCE as high as 18.6%.

Still within the use of push–pull chromophores as interfacial layers, the push–pull nature has also been exploited with layer deposition on top of metal oxides to improve electron injection from the photoactive absorber to the metal oxide, resulting in the enhancement of the device’s photocurrent. For example, Gkini and coworkers [109] have successfully used a bodypi–porphyrin dyad with this aim but with a spin-coated layer. Regarding the key issue of passivation and stability improvement in perovskite cells, some studies have also shown the successful use of push–pull SAMs. For instance, bi-phenyl-based SAMs have been successfully used recently as an interfacial layer between the ZnO electron transport layer and CH_3_NH_3_PbI_3_ hybrid perovskite to improve their stability [110], as shown in Figure 10.

Alagumali and co-workers [111] highlighted the role of push–pull in the passivation of the defects within perovskite active materials. Indeed, they used push–pull D–π–A organic small molecules to passivate the undercoordinated Pb^2+^ defects and to both align the bands and increase hydrophobicity, which results in the improvement in the solar cell stability by hindering the moisture effect (Figure 11).

Still with push–pull compounds attached to the perovskite material, Liu et al. [112] addressed the passivation issue. With this aim, they used 3D polydendate-complexing agents to achieve defect passivation and crystal growth modulation. The 3D complexing agents are phytic acid (PA) and phytate dipotassium (PAD), and the core of the PA material is a six-membered carbon ring surrounded by six phosphate groups, which have been shown to possess 3D structural stability for PA materials. The six branches of the PA material undergo multiple chemical complexations, which result in 3D skeleton templates enabling passivate defects and regulating perovskite crystallization. Another example of passivation is presented in the article by Zhang et al. [113], in which they used polyaromatic molecules based on naphthalene-1,8-dicarboximide (NMI) (Figure 12) and perylene-3,4-dicarboximide (PMI) with different molecular dipoles.

It is shown that such push–pull chromophores provide the passivation of defects and, notably, NMI enables energy band alignment. Particularly, NMI passivation leads to the reduction in grain boundaries (Figure 13) and defect density by about three times, which allows for a reduction in the non-radiative recombination rate and for an increase in the carrier lifetime, thus resulting in an increase of nearly 24% of the perovskite solar cell efficiency (PCE).

Moreover, NMI-modified perovskite cells exhibit noticeably higher stability upon exposure to N_2_ as well as oxygen and humidity (Figure 14).

## 4. SAMs of Push–Pull Chromophores as Dielectric Materials

Self-assembled monolayers of push–pull chromophores are widely used in the design of self-assembled multilayer nanodielectrics (SANDs). Indeed, the structure of such chromophores promotes the electron transfer from the donor to the acceptor moieties through the *p*-conjugated spacer, thus creating a dipole whose strength depends on the nature of the three different parts of the molecule. Being able to assemble these dipoles oriented in the same direction using the self-assembled monolayer strategy enables the generation of dielectric properties in the layers with the dielectric permittivity being higher if the dipoles are strong and well-oriented. The major applications of these SANDs are in the capacitors and gate insulators of OFETs (Organic Field Effect Transistors). In the past several years, Fachetti’s group [114] has been an early leader in the investigation into special types of self-assembled nanodielectrics (SANDs) grown by depositing an alternating σ (Alk) and π (STB) molecular layers with an octochlorotrisiloxane-derived capping layer to stabilize/planarize the assembly and to regenerate a reactive hydroxyl surface for subsequent monolayer deposition. (Figure 15).

From capacitance measurements at 10^2^ Hz, the authors found that the capacitance values depend on the constituent molecules. The higher value was obtained for Type II (710 nF·cm^−2^), while for Type I and III, the values were 400 and 390 nF·cm^−2^, respectively. These results show the importance of the highly polarizable dipolar layer in improving the dielectric constant k, hence increasing the capacitance, which gives SANDs excellent insulating properties. These structures were integrated as gate dielectrics in both p-type and n-type OFETs [115]. The carrier mobilities are comparable to OFETs obtained with SiO_2_ dielectrics but at a lower operating voltage, allowing for a reduction in the power consumption of the device.

Several groups have shown that the turn-on characteristics of Organic Thin-Film Transistors (OTFTs) are related to the dipole moment of the molecule used in the SAM. The permanent dipole feature of push–pull chromophores well-oriented in the SAM on the surface generates the formation of an electrostatic potential, which modulates the densities of carrier charges in the semiconductor channel [116,117,118]. In 2012, Salinas et al. described and correlated the dipole moment of SAM molecules with the threshold voltage of OTFTs [119]. They investigated a set of functionalized n-alkane phosphonic acid molecules with various dipole moments that was deposited on a thin aluminum oxide (Al_2_O_3_) layer to form a hybrid gate dielectric. They showed that the V_th_ is shifted from negative to positive values with the increasing dipole moment of the SAM molecule. Although it was demonstrated that the polarization of the SAM can play an important role in the charge injection into the channel and can therefore impact the V_th_, other parameters like charge trapping or impurities can also affect it. An optimum interface between dielectric and semiconductor is fundamental for efficient device function.

A major part of a SAND studied in the literature was composed of an alternating monolayer of polarized molecules such as phosphonic acid derivatives of stilbazolium salts and high-k dielectric metal oxide (ZnO_2_; HfO_x_) deposited on a metal or semiconductor substrate. These organic–inorganic assemblies provide high gate capacitances, lower gate leakage currents than inorganic film, limited trapped charges, and chemical and thermal stability, as discussed below.

SAND fabrication requires annealing, and such thermal annealing limits SAND compatibility with many plastic or biocompatible substrates and restricts applications such as in biointegrated electronics. This is associated with the densification of the metal oxide layer. In this contribution, the study of the growth, nano structural, and dielectric properties as well as their implementation into the TFTs of zirconium oxide-based SANDs self-assembled using UV radiation processing to make a ZrO_x_ thin film was carried out by Huang et al. [120]. The very high UV photon decomposes the metal oxide precursor and significantly densifies the film (vide infra) [121].

The assembly of the PAE (4-[[4-[bis(2-hydroxyethyl)amino]phenyl]diazenyl]-1-[4-(diethoxyphosphoryl) benzyl]pyridinium bromide) layers between the two ZrO_x_ is known to enhance the stack orientational stability and durability (Figure 16) [115]. To further enhance the insulating properties, the repetition of the PAE-ZrO_x_ bilayer (the self-assembling of PAE in methanol and the UV irradiation of ZrO_x_) deposition n times achieves the fabrication of (Zr-SAND)_n_ superlattices, where n = 1 to 4 (Figure 16b).

An important question in this fabrication process is whether the stilbazolium unit of the PAE layer is affected by exposure to strong UV irradiation. The data from optical absorption evolution suggest that the overlying ZrO_x_ layer stabilizes the PEA layer against UV damage, likely because PAE is not only sandwiched between oxide layers but also chemically locked onto them by the chemistry of the chromophore head and tail. Furthermore, in this way, the PAE layer is also essentially encapsulated and protected from ambient air. The XPS spectra argue that oxide precursor decomposition by UV irradiation is more effective than thermal annealing. The data indicate that upon UV exposure, the ZrO_x_ films become thinner and plausibly denser.

The new UV-irradiated Zr-SANDs (UVD-ZrSANDs) were imaged by AFM to quantify their surface characteristics. All AFM images are essentially featureless, and such exceptionally smooth surfaces are ideal for the fabrication of back-gated transistor devices where even moderate interface roughness can detrimentally affect TFT properties, such as carrier mobility, and illustrate the precise level of control afforded by the present processing methodology. MIS (metal-insulator-semiconductor) sandwich structures containing (UVD-ZrSAND)_n_ layers were fabricated. The results indicate that combining UV-irradiated ZrO_x_ and PAE at room temperature is an efficient route to replace the conventional thermal processing method, thereby realizing high-performance SAND dielectric layers at room temperature.

The molecular dipolar orientation affects the thin-film transistor (TFT) threshold and turn-on voltages for devices based on either p-channel pentacene or n-channel copper perfluorophthalocyanine.

Inverted SANDs are made from inverted PAE units, namely IPAE (Figure 17) [122], which affects the principal OTFT parameters relevant to circuit design and fabrication. Note that although the π-conjugated azastilbazolium cores of PAE and IPAE are identical, there are minor differences in the structures such as a larger distance between the phenylphosphonic acid portion and the core (one atom in PAE and two atoms in IPAE) and, more evidently, two hydroxylethyl fragments in the latter versus one in the former structure. However, the hydroxyethyl group is not the anchoring point of the chromophore to the surface but is simply used to achieve good chemical adhesion to the overlying ZrO_x_ layer. More importantly, it does not drive the self-assembly process, as judged from the kinetics of PAE/IPAE absorption, which are governed by the phosphonic acid fragment and are identical for the two systems.

Even if the main structure of the push–pull chromophore is the pyridinium (+) and aromatic amine (the two different anchors have different impacts on the electronic density along the molecular backbone. The phosphonic acid is electron-withdrawing and aliphatic alcohol is electron-donating. Thus, the push–pull characteristic is more significant in the case of the phosphoric acid being connected to pyridinium and the aliphatic alcohol being connected to the aromatic amine (as is the case of PAE). And for IPAE, the hypsochromic shift is due to the diminished intrinsic electric dipole strength inside the molecule. Specifically, Zr–SAND shifts the threshold and turn-on voltages to more positive values, whereas IZr–SAND shifts them in the opposite direction. Capping these SANDs with –SiMe_3_ groups enhances the effect, affording a 1.3 V difference in turn-on voltage for IZr–SAND vs. Zr–SAND-gated organic TFTs. Such tunability should facilitate the engineering of more complex circuits. This type of junction metal/SAM/dielectric can also be used to modulate the interface thermal conductance.

Lu et al. [123] demonstrated that by using PAE or IPAE chromophores and mixtures of the two as organic linkers between Au and SiO_2_ thin films, the interface thermal conductance of the molecular junction can be tuned based on the relative density of the PAE and IPAE chromophores. The PAE chromophore has two CH_2_CH_2_OH terminal groups compared to one for the IPAE, and these terminal groups control the weak hydrogen bonding between the organic molecule and the Au film. The transition from PAE to IPAE SAMs leads to a 20% decrease in the cross-plane thermal conductance of the junction. Furthermore, the thermal conductance of the mixed PAE–IPAE SAMs (50%:50% molar ratio) is close to a linear combination of the PAE and IPAE SAMs, suggesting that the chromophores act as independent channels for heat conduction.

In the SAND-n samples, on the other hand, we predict that the conductance of the PAE–ZrO_2_ contact is enhanced compared to the conductance of the Au–PAE–SiO_2_ contacts, possibly due to a stronger chemical affinity between the phosphonic acid headgroup and ZrO_2_ compared to SiO_2_ in addition to the stronger adhesion between the hydroxylate tail group with ZrO_2_ versus Au. Although the cross-plane thermal conductance of the SAND-n decreases monotonically with an increasing number of PAE–ZrO_2_ layers, the cross-plane thermal conductivity increases with n. Heat buildup at the organic/inorganic interfaces in the SAND-n resulting from the low thermal conductivity of the ZrO_2_ layers and the interface thermal resistance at the PAE–ZrO_2_ interface can lead to increased temperatures in these films beyond their suitable operational limits, leading to thermoelastic strain and the reconfiguration of the PAE molecules.

A new hafnium oxide–organic self-assembled nanodielectric (Hf-SAND) material consisting of regular, alternating π-electron layers of 4-[4-[bis(2-hydroxyethyl)amino]phenyl]diazenyl]-1-[4-(diethoxyphosphoryl) benzyl]pyridinium bromide) and HfO_2_ nanolayers is reported in [124]. The goal of this research is to develop enhanced-performance hybrid superlattice dielectrics using alternative oxides as the SAND oxide component. The motivation for extending to hafnia is based on reports indicating the differential affinity of phosphonic acids for various oxides versus ZrO_2_, along with HfO_2_ thermodynamic and surface chemical differences that may beneficially affect the dielectric properties at low process temperatures [125].

Figure 18 shows the fabrication scheme for the new Hf-SAND-n films. Multilayer variants can be prepared by repeating the indicated self-assembly steps in an iterative fashion, where the n index indicates the number of π-electron/HfO_x_ bilayers grown on top of the initial HfO_x_ priming layer. The resulting HfO_x_/π-electron bilayer is then “capped” with a second layer of HfO_x_ by spin-coating and baking. This process regenerates the metal oxide surface for additional layers (if desired) of phosphonic acid PAE SAM to initiate the next nanodielectric repeat unit.

Hf-SAND-n variants ranging from one to four bilayers (Hf-SAND-1∼4) were imaged by AFM to quantify the surface roughness, conformality, and contiguity. These films exhibited RMS (Root Mean Square) roughness values ranging from 1.3 Å for a single layer to 1.7 Å for the four-layer variant, highlighting a negligible additional roughness compared to the native Si oxide surface [126]. This modest increase in roughness is consistent with the deposition of additional dielectric layers. The exceptionally smooth surfaces are ideal for the fabrication of back-gated transistor devices where even moderate interface roughness can detrimentally affect TFT properties [127]. To analyze the elemental and chemical composition of the completed Hf-SAND multilayers, XPS was employed. It was concluded that the O 1s signal observed after the present 150 °C processing is qualitatively like the metal oxide spectra of samples processed at much higher temperatures (300 °C) [128].

In all electronic circuits, it is critical to limit TFT gate dielectric leakage currents for efficient switching and to minimize power consumption during device operation. In the case of the present capacitor structures, Hf-SAND-n dielectric layers’ leakage is several orders of magnitude lower than that of native SiO_2_ capacitors (1 A/cm^2^ at ±2 MV/cm) and is comparable to previous reports utilizing either solution phase self-assembly or vacuum deposition dielectric growth techniques, such as ALD [129], which typically afford optimized leakage current densities of ∼10^−8^ A/cm^2^ or less. The thicker, multilayer Hf-SAND variants exhibit significantly lower electric field normalized leakages, which should allow larger voltage biasing windows, useful for several transistor applications. A single layer of Hf-SAND can achieve a capacitance density greater than 1 μF/cm^2^ (1.1 μF/cm^2^ measured) versus 0.75 μF/cm^2^ for a single layer of Zr-SAND and 0.71 μF/cm^2^ for the silane-based SAND type II structure reported in the literature [114]. This represents a capacitance enhancement of nearly 50% over current-generation SAND materials and enables microfarad capacitance densities for the first time from a solid-state SAM–metal oxide hybrid dielectric, which is processable at low temperatures in an ambient atmosphere. In the literature, a study using ZrP as the solid support for the PAE SAM showed 700 nF/cm^2^ for 1*p*-SAND and 520 nF/cm^2^ for 2*p*-SAND in the accumulation regime (0 to +1.0 V). Figure 19 graphically illustrates Hf-SAND dielectric trends in terms of capacitance density versus the bilayer n number, inverse capacitance versus n, EOT, and the overall dielectric permittivity (k) of Hf-SAND versus thickness. Also note that as n increases, the overall dielectric constant (k_eff_) also increases. This implies that the PAE layer, which is the majority component in subsequent layers, has a larger k than that of the inorganic oxide. This supports the conclusion of Yoon et al. [114] that the stilbazolium group has very high permittivity, generally greater than what can normally be achieved with low-temperature sol–gel oxides.

Recently, this type of SAND has been deposited on an IGZO semiconductor as the underlying channel layer. This device exhibits impressive electron mobility (µsat = 19.4 cm^2^ V^−1^ s^−1^) and low threshold voltage (V_th_ = 0.83 V) compared to a similar device without a push–pull layer combined with Hf oxide [130].

## 5. Various Electrical and Photonic Properties Generated by Push–Pull Chromophore Assemblies

The ordered assembly of push–pull chromophores at the surface is mandatory to settle the expected right functionalization property. Obviously, the nature of the linker between the chromophore and the surface plays a key-role in the organization. Hupfer and coworkers [131] have studied the role of aryl versus alkyl linker on the supramolecular structure and the optoelectronic properties of tripodal push–pull thiazoles. Despite its more insulating property, the alkyl linker has been found to give higher conductivity to the assembly, presumably because it promotes more degrees of freedom, enabling supramolecular rearrangement upon electrical measurement. There are very few studies on push–pull electrical properties within single push–pull molecule thick junctions [132]. Some works dealing with single push–pull junctions are noticeable, such as the use of mechanically controlled break junction to study the electrical conductance modification activated by an external electric field, such as resonance features with oligo(phenyleneethynylene) wires with donor–acceptor substitution on the central ring [133], or the electrical bistability of Fe^II^-bis-terpyridine push–pull complexes activated by an external electric field which triggers a spin crossover transition [134] (Figure 20).

In the literature, self-assembled push–pull chromophores are mainly used to form supramolecular ordered layers. For instance, Li et al. [135] have processed the layer-by-layer stacking of self-assembled push–pull derivatives to form 2D organic crystals by layering amphiphilic-like stacking with alternating attractive layers (AL) and repulsive layers (RL) to build supramolecular “push–pull” assemblies within a liquid surface-assisted solution self-assembly strategy (Figure 21). A monolayer (~1.5 nm thick) is made of two outside RLs and a sandwiched AL with such high packing density that, interestingly, the stacks exhibit outstanding photoelectric integrated properties, with high mobility, a high crystalline state, and superior deep-blue laser characteristics.

Organization-dependent photoelectric properties have also been studied on intermolecular charge transfer push–pull derivatives, such as strong dithiole–nitrofluorene push–pull dyads [136]. On such dyad assemblies controlled by symmetrical or asymmetrical dipole–dipole interaction, the authors have reported aggregation-induced emission and particularly the red-emitting behavior of the well-formed hierarchical micro- and nanostructures, which could find interest in OLEDs. Aggregation-induced emission enhancement and strong intermolecular charge transfer have also been used within a solution-processed non-doped orange-red-emitting multifunctional organic fluorophore made of two terminal attachments of a push–pull moiety separated by a biphenyl free rotor and its copolymers (Figure 22) [137]. Still controlling the interaction, Kim and co-workers have developed a 2D single-crystal down to two monolayers of (2E,2′E)-3,3′-(2,5-difluoro-1,4-phenylene) bis (2-(5-(4-(trifluoromethyl)phenyl) thiophen-2-yl)acrylonitrile), which exhibit field-effect electron mobility and photoresponsivity of 3.6 × 10^3^ A W^−1^ under green-light-emitting diode irradiation [138]. Such results constitute the first example of green-sensitive 2D organic phototransistors. All these studies point out the role of the organization within self-assembled push–pull multilayers that could be profitably further developed within a self-assembled monolayer strategy.

## 6. Conclusions

In this paper, we have reviewed the beneficial use of organized layers of push–pull chromophores to generate photoelectrical and electrical properties, mainly focusing on the self-assembled monolayer strategy to arrange push–pull assemblies. Regarding electrical properties, self-assembled monolayers have mostly been devoted to high-k dielectric thin layers for possible applications in nanoelectronic devices such as nanotransistors. Push–pull assemblies have also been extensively studied for improving DSSC and perovskite solar cell operation. Within DSSC push–pull assemblies, they could behave as efficient sensitizers, whereas for perovskite solar cells, they have been shown to be able to address the critical issues of energy band alignment, improvement in the electron transfer at the interface, and the increase in stability as a passivating interfacial layer. At last, some other photo-electrical properties such as high mobility, high crystalline state, superior deep-blue laser characteristics, bistability, and aggregation-induced emission, particularly red-emitting behavior, finding interest in OLEDs or green-sensitive 2D organic phototransistors, have also been shown to arise from ordered assemblies of push–pull chromophores. In all these studies, the intrinsic push–pull properties are of crucial importance. Indeed, acceptor, donor, and spacer natures must be carefully chosen to tune the desired properties. For example, to achieve high dielectric properties within SANDs, the strength of the acceptor and the donor must be maximized as well as the spacer electron transfer rate. Within DSSC, the moieties should promote a strong and wide molar spectral absorption (e.g., using porphyrins and phthalocyanines) and a high intramolecular charge transfer rate (spacer). Moreover, the right design of push–pull chromophores through the nature and length of the π-bridge groups, together with using various acceptor and donor groups with different electron affinities, enables the development of the energy-level engineering of chromophores on metal oxide surfaces. In interfacial layers, the moieties are chosen to match the energy levels. Furthermore, their organization appeared to play a key role in generating the desired properties. For instance, in DSSC applications, the dyes should be grafted onto the inorganic mesoporous semiconductor oxide film in controlled organized stacks, thanks to an adequate anchoring group and avoiding aggregation, in order to allow high charge-transfer kinetics. For this purpose, various parameters could be optimized such as introducing a flexible linker between the chromophore and the surface and the use of non-charged push–pull chromophores in order to promote an organized packed assembly.

## Figures and Tables

**Figure 1 molecules-29-00559-f001:**
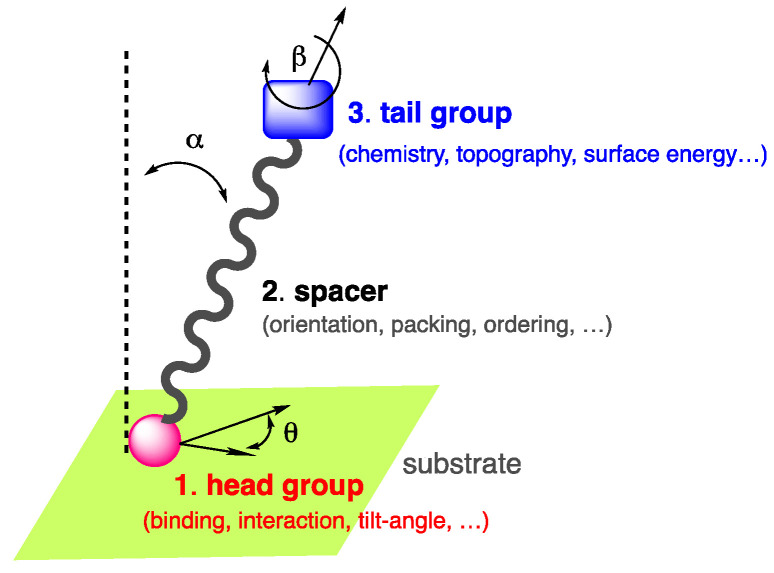
Amphiphilic self-assembling molecule showing the end group, spacer, and tail group in interaction with a substrate.

**Figure 2 molecules-29-00559-f002:**
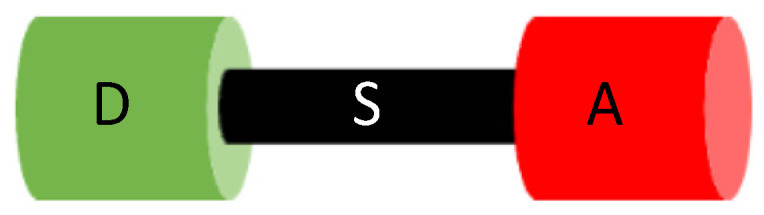
Push–pull π-conjugated chromophore structure: D—electron-donating unit (push); A—electron-withdrawing unit (pull); S—conjugated bridge (spacer).

**Figure 3 molecules-29-00559-f003:**
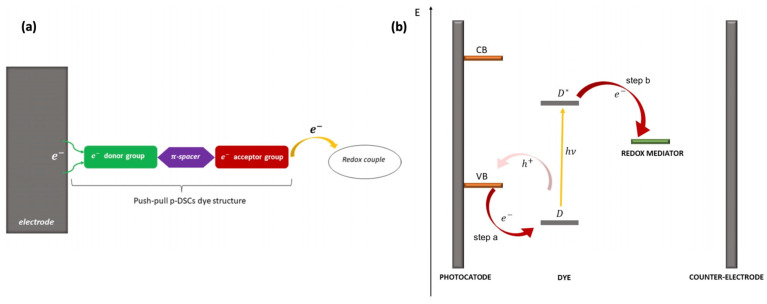
(**a**) Structure of *p*-DSSC dye (green: electron donor; violet: π-bridge; red: electron acceptor), (**b**) electron pathway in an illuminated *p*-DSSC. From reference [66]. No permission required.

**Figure 4 molecules-29-00559-f004:**
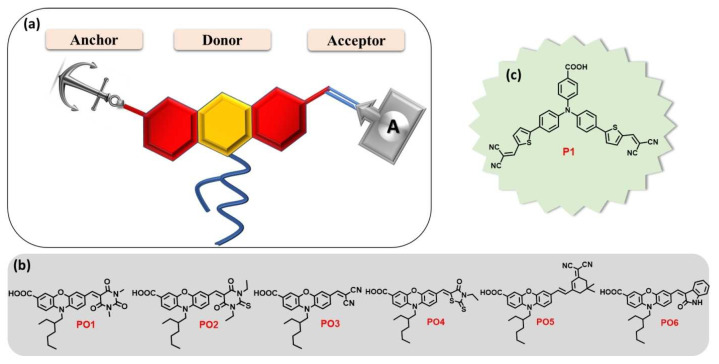
(**a**) Molecular design of new D-A dyes PO1-6 whose studied structures with a carboxylic acid anchoring group and different acceptors are presented in (**b**). The benchmark reference dye P1 is shown in (**c**). Reprinted with permission from reference [72]. Copyright (2024) John Wiley and Sons. Further permissions related to the material excerpted should be directed to John Wiley and Sons.

**Figure 5 molecules-29-00559-f005:**
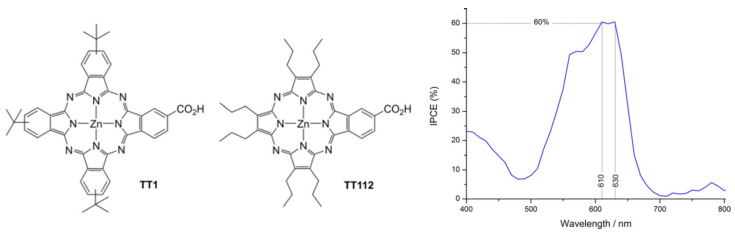
Chemical structures of Pc TT1 and Pz TT112 porphyrazine-based sensitizers and IPCE (incident photon-to-current efficiency) measured with TT112. Reprinted with permission from reference [86]. Copyright (2024) John Wiley and Sons. Further permissions related to the material excerpted should be directed to John Wiley and Sons.

**Figure 6 molecules-29-00559-f006:**
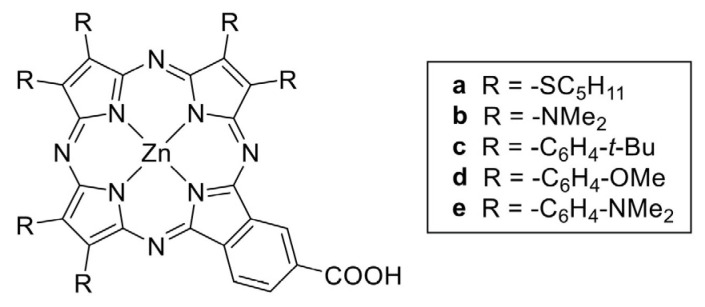
Chemical structures of A3B panchromatic push–pull porphyrazines studied in reference [87], still incorporating the electron-withdrawing B moiety of TT112 presented in Figure 5 but with subunits made of pyrrole rings substituted with different donor groups. No permission required.

**Figure 7 molecules-29-00559-f007:**
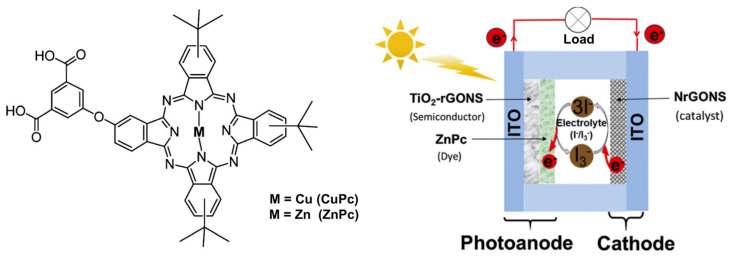
Asymmetric synthesized CuPc and the scheme of a phthalocyanine (ZnPc)-based DSCC. Reprinted with permission from reference [91]. Copyright (2024) Elsevier. Further permissions related to the material excerpted should be directed to Elsevier.

**Figure 8 molecules-29-00559-f008:**
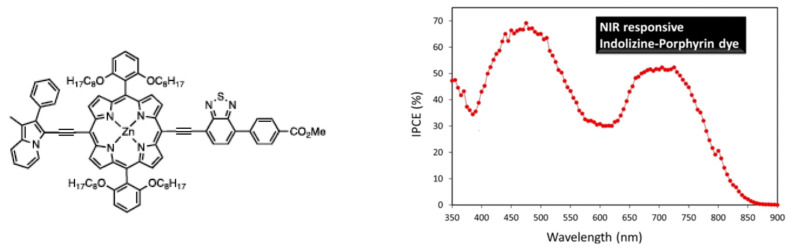
Indolizine–porphyrin push–pull dye and evidence of its absorption in the near-infrared region. Reprinted with permission from reference [106]. Copyright (2024) American Chemical Society. Further permissions related to the material excerpted should be directed to the American Chemical Society.

**Figure 9 molecules-29-00559-f009:**
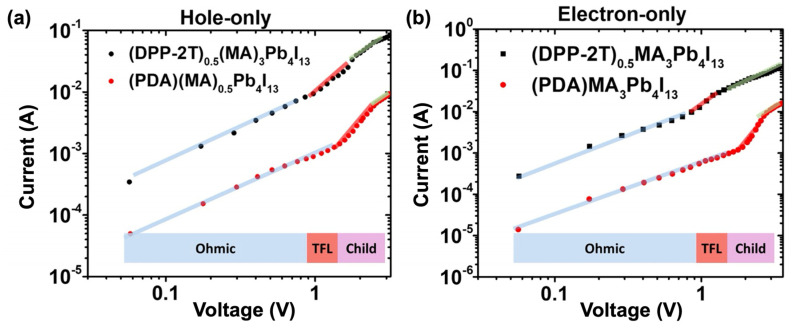
Hole (**a**) and electron (**b**) current measured on perovskite solar cell showing higher values for the cell incorporating DPP-2T push–pull chromophores (blue line = ohmic, red line = TFL and green line = child). Reprinted with permission from reference [108]. Copyright (2024) Elsevier. Further permissions related to the material excerpted should be directed to Elsevier.

**Figure 10 molecules-29-00559-f010:**
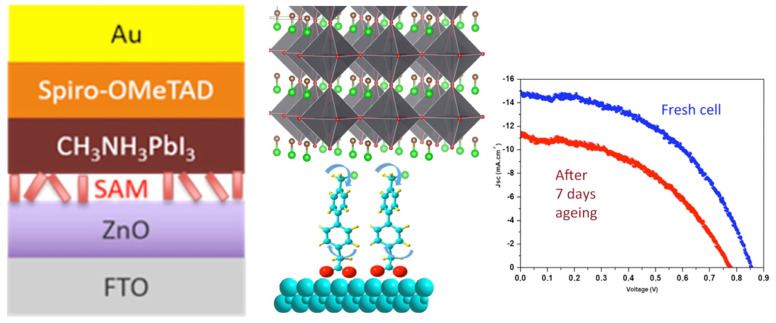
Interfacial SAM between the ZnO electron transport layer and perovskite (**left**) and their effect on the solar cell stability (**right**). Reprinted with permission from reference [110]. Copyright (2024) American Chemical Society. Further permissions related to the material excerpted should be directed to American Chemical Society.

**Figure 11 molecules-29-00559-f011:**
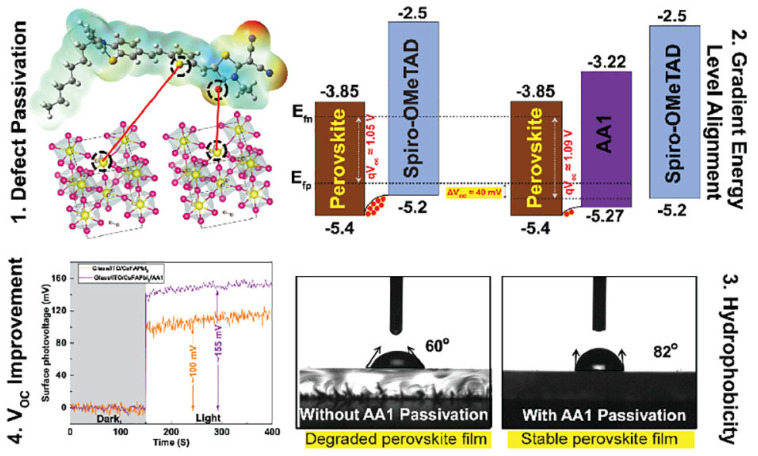
Scheme of the push–pull incorporation within perovskite materials, induced band alignment, and evidence of the increase in open circuit voltage and hydrophobicity. Reprinted with permission from reference [111]. Copyright (2024) American Chemical Society. Further permissions related to the material excerpted should be directed to American Chemical Society.

**Figure 12 molecules-29-00559-f012:**
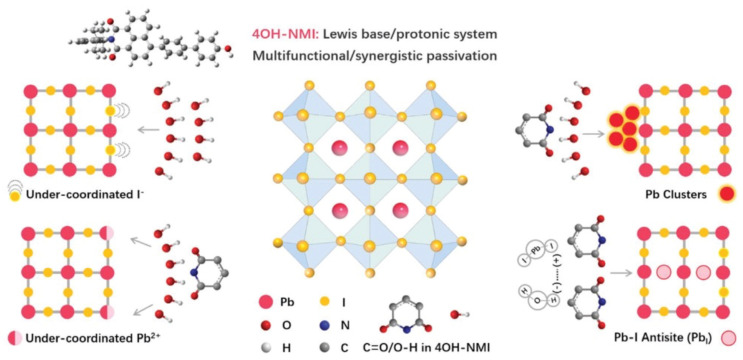
Scheme of the different chemical passivation functions of 4OH-NMI with corresponding defects in perovskite. Reprinted with permission from reference [113]. Copyright (2024) John Wiley and Sons. Further permissions related to the material excerpted should be directed to John Wiley and Sons.

**Figure 13 molecules-29-00559-f013:**
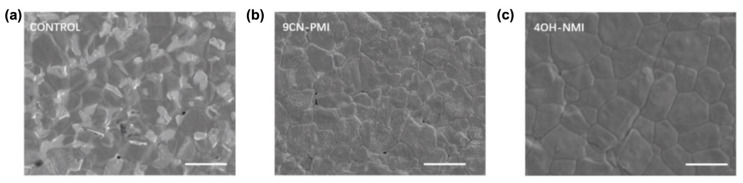
Thermal admittance spectroscopy images showing the higher grain boundary reductions for NMI-modified than for PMI-modified perovskite solar cells in comparison to the non-modified cell (control image). Scanning electron microscopy (SEM) images: (**a**) control film; (**b**) 9CN-PMI -modified film and (**c**) 4OH-NMI-modified film. The scale is 2 μm. Reprinted with permission from reference [113]. Copyright (2024) John Wiley and Sons. Further permissions related to the material excerpted should be directed to John Wiley and Sons.

**Figure 14 molecules-29-00559-f014:**
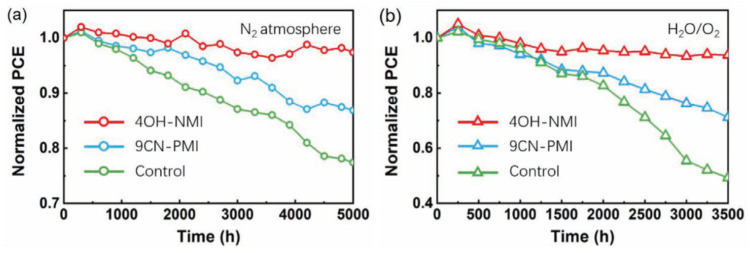
Stability test of 4OH-MNI-modified (red), 9CN-PMI-modified (blue) and control (green) perovskite solar cells upon exposure (**a**) to N_2_ atmosphere (**b**) oxygen, and 20–30% relative humidity. Reprinted with permission from reference [113]. Copyright (2024) John Wiley and Sons. Further permissions related to the material excerpted should be directed to John Wiley and Sons.

**Figure 15 molecules-29-00559-f015:**
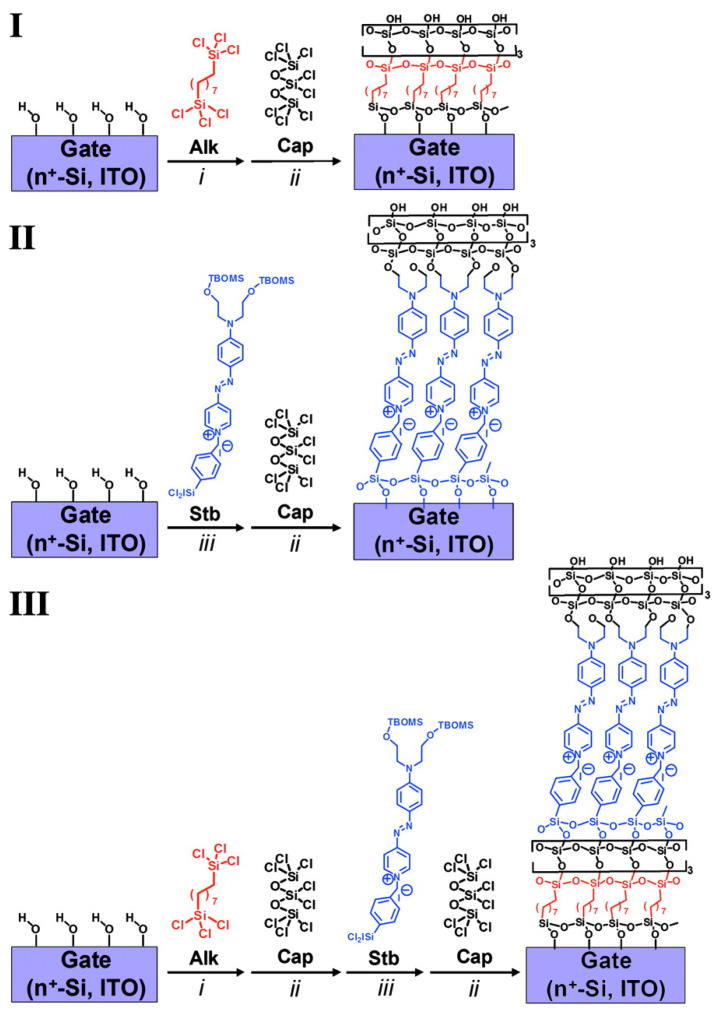
Schematic of three different self-assembly of nanodielectrics (**I**–**III**) on highly n-type doped Si(100) wafers with a 1.5 nm native oxide or smooth ITO as substrate/gate electrodes. Nanodielectric layers were sequentially deposited from solutions of silane precursors Alk, Stb, or Cap (conditions were as follow *i*: 5 mM Alk in dry toluene at 0 °C in N_2_ for 1h; *ii*: 34 mM Cap in dry pentane at room temperature in N_2_ for 25 min.; or *iii*: 10 mM Stb in dry tetrahydrofuran at 60 °C in N_2_ for 15 min. followed by hydrolysis with acetone-H_2_O solution). Reprinted with permission from reference [114]. Copyright (2024) National Academy of Sciences. Further permissions related to the material excerpted should be directed to National Academy of Sciences.

**Figure 16 molecules-29-00559-f016:**
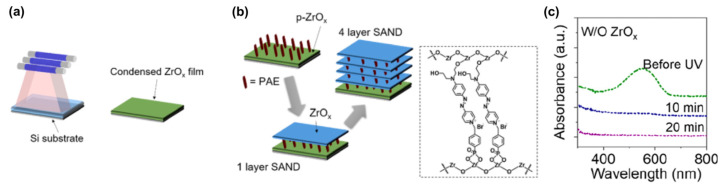
(**a**) Schematic representation of ZrO_x_ film fabrication by UV irradiation. (**b**) Fabrication procedure for UV-densified zirconia self-assembled nanodielectric (UVD-Zr-SAND)_n_ multilayers and molecular structure of the PEI organic component of SAND. (**c**) Absorbance spectra of a primer-ZrO_x_-PAE film without the ZrO_x_ capping layer before and after irradiation. Reprinted with permission from reference [120]. Copyright (2024) American Chemical Society. Further permissions related to the material excerpted should be directed to American Chemical Society.

**Figure 17 molecules-29-00559-f017:**
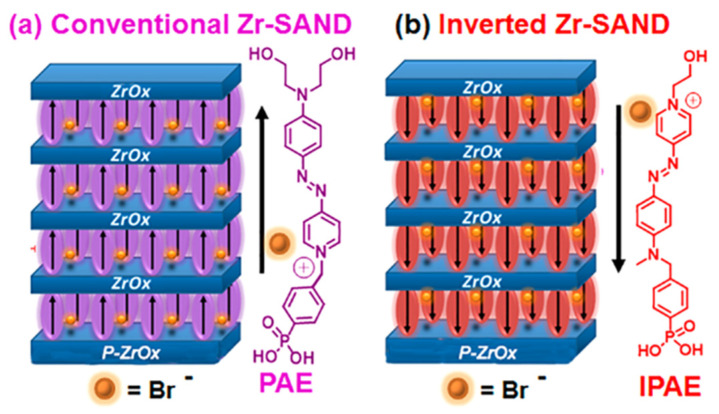
Dielectric stacks comprising four-chromophore/ZrO_x_ layers on top of the ZrO_x_ (*p*-ZrO_x_) primer film. (**a**) Conventional Zr-SAND with a phosphonate π-electron (PAE) unit. (**b**) Inverted IZr-SAND with an inverted PAE (IPAE) π-unit. Reprinted with permission from reference [122]. Copyright (2024) American Chemical Society. Further permissions related to the material excerpted should be directed to American Chemical Society.

**Figure 18 molecules-29-00559-f018:**
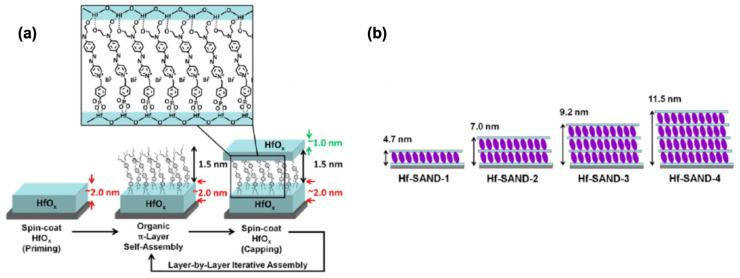
(**a**) Solution-based Hf-SAND self-assembly procedure employed in this study. (**b**) Schematic of the various Hf-SAND multilayers produced including the corresponding X-ray reflectivity-derived thicknesses (Hf-SAND-1, -4) and estimated thicknesses (HfSAND-2, -3). Reprinted with permission from reference [124]. Copyright (2024) American Chemical Society. Further permissions related to the material excerpted should be directed to American Chemical Society.

**Figure 19 molecules-29-00559-f019:**
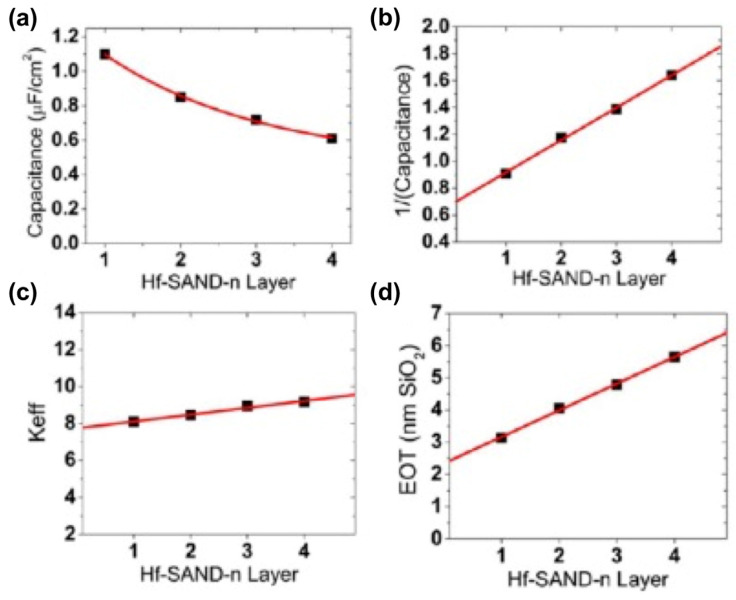
Plots with the best fits of Hf-SAND-n MIS properties. (**a**) Assuming 1/x decay dependence of capacitance density versus Hf SAND layer number n. (**b**) Inverse capacitance versus layer number linear relationship. (**c**) Increasing effective dielectric constant k_eff_ versus layer number n. (**d**) EOT versus layer number n. Reprinted with permission from reference [124]. Copyright (2024) American Chemical Society. Further permissions related to the material excerpted should be directed to American Chemical Society.

**Figure 20 molecules-29-00559-f020:**
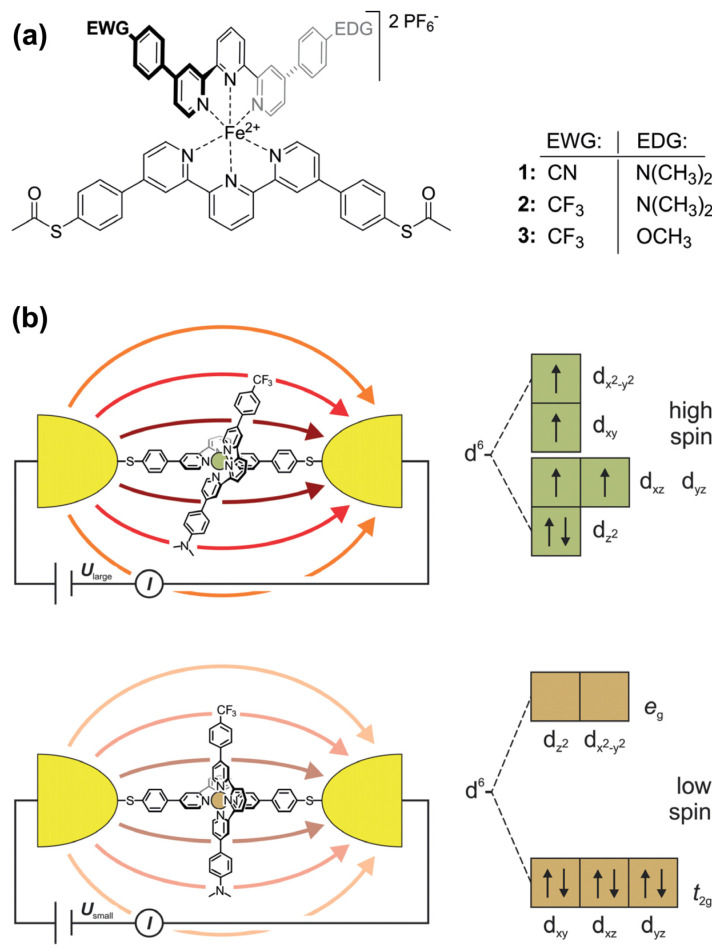
(**a**) Fe^II^-bis-terpyridine complexes 1–3 as E-field-sensitive spin crossover switching junctions. (**b**) Scheme of the E-field-triggered switching using the distortion of the Fe^II^-pyridine ligand sphere upon the application of a large enough voltage. From reference [134]. No permission required.

**Figure 21 molecules-29-00559-f021:**
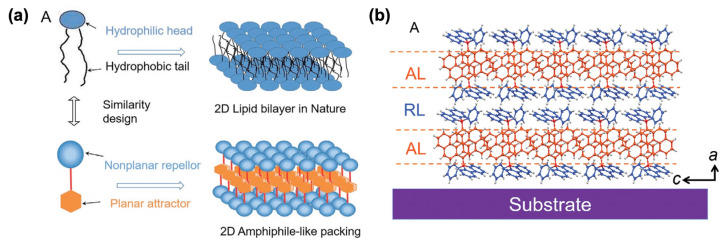
Schematic representation of the 2D amphiphile-like packing layer similar to a 2D lipid bilayer in nature (**a**) and the AL/RL alternate stacks developed (**b**). Reprinted with permission from reference [135]. Copyright (2024) John Wiley and Sons. Further permissions related to the material excerpted should be directed to John Wiley and Sons.

**Figure 22 molecules-29-00559-f022:**
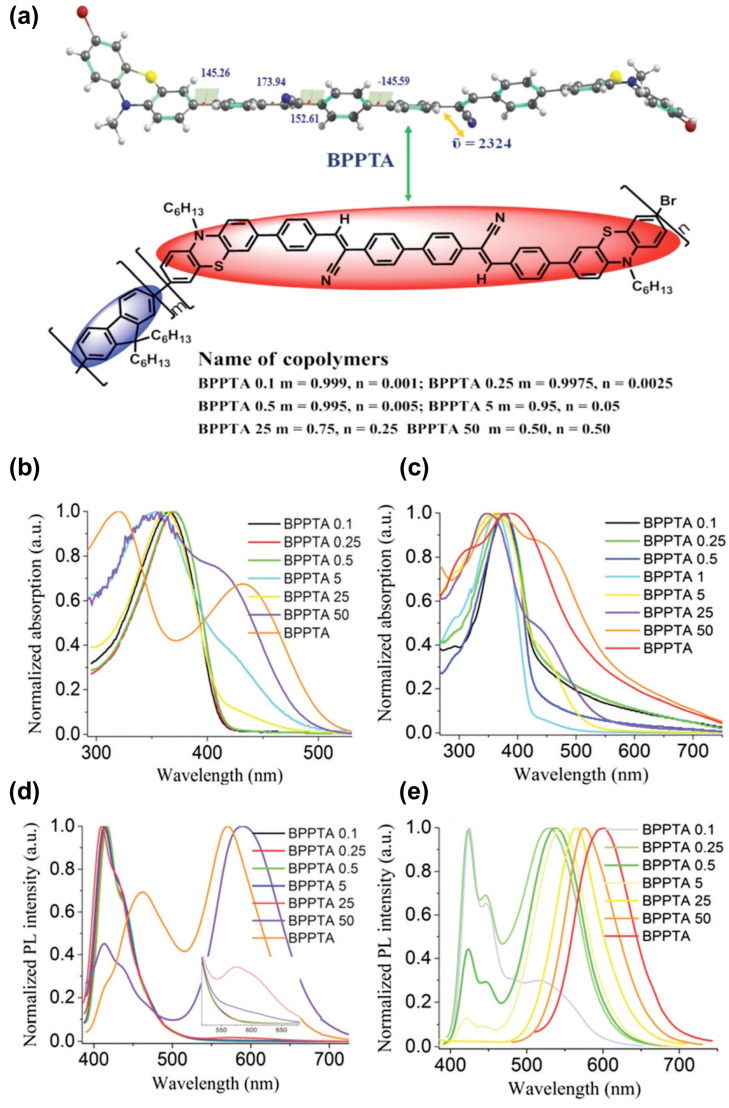
(**a**) Optimized structure of BPPTA and various molecular compositions of BPPTA–fluorene copolymers. Absorption spectra in solution (**b**) and as thin films (**c**) of BPPTA and its copolymers. Photoluminescence (PL) spectra in solution (**d**) solution and as thin films (**e**) of the copolymers and BPPTA under irradiation at ~360 nm. Reprinted with permission from reference [137]. Copyright (2024) Royal Society of Chemistry. Further permissions related to the material excerpted should be directed to the Royal Society of Chemistry.

## Data Availability

Not applicable.

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
