# Peer review of "Self-Assembled Monolayers of Push–Pull Chromophores as Active Layers and Their Applications"

_molecules, 2024, doi:10.3390/molecules29030559_

Round 1

Reviewer 1 Report

Comments and Suggestions for Authors

This review summarized the recent advance in applications of self-assembled monolayers of push-pull chromophores as active layers, with special attentions on dye-sensitized solar cell applications, improving perovskite solar cell performances, and functioning as dielectric materials. This review is helpful to newcomers to the related fields. I recommend its acceptance for publication. However, the following issues should be well addressed before its acceptance.     

1. In keywords, ‘self-assembled and ‘self-assembly’ are duplicates, please delete either of them.  

2. In Line 322 on Page 10, please give the full name here when PAE is first introduced, just like in caption of Figure 16. Likewise, please give the full names for MIS in Line 346 on Page 11 and RMS in Line 423 on Page 13, respectively.

3. There are some errors in typewriting, for example, ‘Yoon et al [114].’ in Line 454 on Page 14. Please carefully check the manuscript.

4. As an outlook, how to obtain SAMs of push-pull chromophores with desirable properties? What are the challenges?

5. How to design the suitable push-pull chromophores for particular applications like DSSC? What are the crucial points?

6. Are the electro-optical properties of SAMs of push-pull chromophores largely dependent on push-pull chromophores themselves or on how to assemble?

Author Response

We thanks the reviewer for his valuable comments and we have provided a reply for each point.

  1. In keywords, ‘self-assembled’ and ‘self-assembly’ are duplicates, please delete either of them.  

Self-assembly has been removed accordingly.  

  1. In Line 322 on Page 10, please give the full name here when PAE is first introduced, just like in caption of Figure 16. Likewise, please give the full names for MIS in Line 346 on Page 11 and RMS in Line 423 on Page 13, respectively.

Full names of the acronyms are provided in the body text when first introduced. 

PAE (4-[[4-[bis(2-hydroxyethyl)amino]phenyl]diazenyl]-1-[4-(diethoxyphosphoryl) benzyl]pyridinium bromide) 

MIS (metal-insulator-semiconductor) 

RMS (Root Mean Square) roughness 

  1. There are some errors in typewriting, for example, ‘Yoon et al [114].’ in Line 454 on Page 14. Please carefully check the manuscript.

The MS has been checked and the typewriting errors have been corrected. 

  1. As an outlook, how to obtain SAMs of push-pull chromophores with desirable properties? What are the challenges?

We thank the Referee for his smart request which should help to improve the understanding of the challenges to address in push-pull design and assembly. We have therefore slightly modified the conclusion adding the following paragraph in order to address such outlook: 

“ In all these studies, the intrinsic push-pull properties are of crucial importance. Indeed, acceptor, donor and spacer natures must be carefully chosen to tune the desired properties. For example, to achieve high dielectric properties within SAND, the strength of the acceptor and the donor must be maximized as well as the spacer electron transfer rate. Within DSSC, the moieties should promote a strong and wide molar spectral absorption (e.g. using porphyrins and phthalocyanines) and a high intramolecular charge transfer rate (spacer). Moreover, the right design of push-pull chromophore through the nature and length of the π -bridge groups, together with using various acceptor and donor groups with different electron affinities, enables to develop energy-level engineering of chromophores on the metal oxide surface. In interfacial layers, the moieties will be chosen to match the energetic levels. Beside, their organization appeared to play a key-point to generate the desired properties. For instance, in DSSC application the dyes should be grafted onto the inorganic mesoporous semiconductor oxide film in controlled organized stacks, thanks to an adequate anchoring group and with avoiding aggregation, in order to allow high charge-transfer kinetics. For this purpose various parameters could be optimized such as introducing a flexible linker between the chromophore and the surface, and to use non-charged push-pull chromophores in order to promote an organized packed assembly.” 

  1. How to design the suitable push-pull chromophores for particular applications like DSSC? What are the crucial points?

As highlighted in the various articles reviewed in the manuscript, various points have to be taken into account in the push-pull design and assembly in order to optimize DSSC performances :  

  • first, they should exhibit a strong and wide molar spectral absorption (for instance it could be enhanced by the use or porphyrins or phthalocyanines) 
  • a peculiar attention should be given to the enhancement of the intramolecular charge transfer rate mainly through the right choice of the π-conjugated spacer 
  • it is necessary to develop energy-level engineering of chromophores on the metal oxide surface, which can be achieved by the right design of push-pull chromophore through the nature and length of the π -bridge groups, together with using various acceptor and donor groups with different electron affinities 
  • the dyes should be grafted onto the inorganic mesoporous semiconductor oxide film in controlled organized stacks, thanks to an adequate anchoring group and with avoiding aggregation, in order to allow high charge-transfer kinetics; for the latter, using a π-conjugated spacer between the chromophore and the anchoring group is a way to enhance electron transfer. 

A summary of these points have been added in the conclusion meanwhile the request 4 have been addressed (see above). 

  1. Are the electro-optical properties of SAMs of push-pull chromophores largely dependent on push-pull chromophores themselves or on how to assemble?

Both SAMs properties will depend on intrinsic properties of the push-pull chromophores and how there are arranged within the monolayer.  

Reviewer 2 Report

Comments and Suggestions for Authors

In this manuscript, J. Wang et al. summarize the recent progresses in SAMs of push-pull chromophores and their applications in solar cells and functional devices. This review is of interest to a broader audience and thus suitable for publication in Molecules. Before acceptance, some minor concerns need to be addressed (see comments attached).

Author Response

Please find enclosed our replies point-by-point. We thanks the reviewer for his remarks and comments.

  1. In Section 1, the authors give a detailed introduction of SAMs whereas not specifically on that composed of push-pull chromophores. It may be better to shorten the intro while put more emphasis on push-pull chromophores.

To our opinion it is necessary to well describe the parameters impacting the SAM organization since it is of prime importance to generate the properties within push-pull assemblies. Since such parameters have been mostly studied on other type of molecules a long part of the section has necessarily been devoted to the study of non-push-pull molecules before push-pull chromophores have been finally addressed in the last paragraph 

  1. A clear definition of push-pull chromophores is lacking. A brief explanation (either in text or schematic illustration) would benefit especially for those who are not familiar with this topic. 

A paragraph has been added to give a brief explanation about push-pull chromophores. 

  1. The authors may want to add a brief description of the operating principles shown in Fig. 2. 

The description of the operating principle schemed in the former Fig.2 (now Fig. 3) has been added in the text as follows:  

“Upon light absorption, an electron is transferred from a p-type semiconducting substrate (valence band VB) to the donor group of the dye (D) which becomes excited (D*), and the acceptor moiety withdraws the electron from the donor through the spacer (step a). The latter both acts as an electron vehicle and is the seat of charge separation. At last, the acceptor donates the electron to the oxidized form of the redox mediator (step b)." 

  1. In line 286, the meaning of ‘these’ is not clear. 

Yes indeed. We have replaced “these” by “capacitance values”, as follows: 

«â€¯From capacitance measurements at 102 Hz, they found that capacitance values depend on the constituent molecules. » 

  1. In Section 4, the role of push-pull chromophores is not emphasized. Consider shortening the “fabrication” part and discussing more on the electronic properties of the SAMs of push-pull chromophores. 

The referee’s request is praiseworthy in order to highlight the specific role of push-pull, but we think that the fabrication details are appropriate to support the dipole generation and orientation to explain the expected dielectric properties. However, it is right that the electronic properties of the push-pull chromophores has somehow not been well described and we have added some explanation on the push-pull role in the beginning of the section 4, as follows :  

«â€¯Indeed, the structure of such chromophore promotes an electron transfer from the donor to the acceptor moieties through the π-conjugated spacer, thus creating a dipole whose strength depends on the nature of the three different parts of the molecule. Being able to assemble these dipoles oriented in the same direction using self-assembled monolayer strategy enables to generate dielectric properties in the layers with dielectric permittivity even higher that the dipoles are strong and well-oriented. »  

  1. More references are needed. For instance, line 103, 313, etc. 

Concerning line 103 the reference must be found at the end of the sentence, i.e. [60]. 

For the line 313, the sentence is an introduction to the following paragraphs where references can be found, and we have therefore added «â€¯as discussed below » at the end of the sentence. 

  1. Abbreviations should be defined at first appearance. For instance, OFET in line 278, PAE in line 322, etc. 

For all abbreviations a description is given when appeared first in the body text